# Particulate Matter 2.5 Mediates Cutaneous Cellular Injury by Inducing Mitochondria-Associated Endoplasmic Reticulum Stress: Protective Effects of Ginsenoside Rb1

**DOI:** 10.3390/antiox8090383

**Published:** 2019-09-09

**Authors:** Mei Jing Piao, Kyoung Ah Kang, Ao Xuan Zhen, Pincha Devage Sameera Madushan Fernando, Mee Jung Ahn, Young Sang Koh, Hee Kyoung Kang, Joo Mi Yi, Yung Hyun Choi, Jin Won Hyun

**Affiliations:** 1Jeju Research Center for Natural Medicine, Jeju National University School of Medicine, Jeju 63243, Korea; 2Laboratory of Veterinary Anatomy, College of Veterinary Medicine, Jeju National University, Jeju 63243, Korea; 3Department of Microbiology and Immunology, Inje University College of Medicine, Busan 47392, Korea; 4Department of Biochemistry, College of Oriental Medicine, Dongeui University, Busan 47340, Korea

**Keywords:** particulate matter 2.5, ginsenoside Rb1, endoplasmic reticulum stress, oxidative stress, apoptosis

## Abstract

The prevalence of fine particulate matter-induced harm to the human body is increasing daily. The aim of this study was to elucidate the mechanism by which particulate matter 2.5 (PM_2.5_) induces damage in human HaCaT keratinocytes and normal human dermal fibroblasts, and to evaluate the preventive capacity of the ginsenoside Rb1. PM_2.5_ induced oxidative stress by increasing the production of reactive oxygen species, leading to DNA damage, lipid peroxidation, and protein carbonylation; this effect was inhibited by ginsenoside Rb1. Through gene silencing of endoplasmic reticulum (ER) stress-related genes such as *PERK*, *IRE1*, *ATF*, and *CHOP*, and through the use of the ER stress inhibitor tauroursodeoxycholic acid (TUDCA), it was demonstrated that PM_2.5_-induced ER stress also causes apoptosis and ultimately leads to cell death; however, this phenomenon was reversed by ginsenoside Rb1. We also found that TUDCA partially restored the production of ATP that was inhibited by PM_2.5_, and its recovery ability was significantly higher than that of ginsenoside Rb1, indicating that the process of ER stress leading to cell damage may also occur via the mitochondrial pathway. We concluded that ER stress acts alone or via the mitochondrial pathway in the induction of cell damage by PM_2.5_, and that ginsenoside Rb1 blocks this process. Ginsenoside Rb1 shows potential for use in skin care products to protect the skin against damage by fine particles.

## 1. Introduction

Particulate matter 2.5 (PM_2.5_), a major cause of air pollution, is a particle with an aerodynamic size of 2.5 microns or less. Although increasing studies have evaluated the effects of PM_2.5_ exposure on human health, most studies have focused on respiratory diseases such as respiratory inflammation, asthma, and cardiovascular diseases [1,2,3]. Owing to increasingly severe air pollution, the incidence of skin diseases has also increased. Ambient air pollution is considered an important risk factor for skin diseases. Diesel exhaust components containing PM_2.5_ include black carbon, hydrocarbons (C_14_–C_35_), heterocyclic polycyclic aromatic hydrocarbons (PAHs), and their derivatives [4]. The highly lipophilic PAHs contained in PM_2.5_ are major factors leading to mutagenicity [5] and easily penetrate the skin [6]. The aromatic hydrocarbon receptor (AhR) is activated by PM in the epidermis [7]. Upon activation by PAHs, AhR may increase the production of intracellular reactive oxygen species (ROS) [8]. PM_2.5_ also has a detrimental effect on human keratinocytes [9,10]. We previously reported the mechanism underlying damage to skin cells via PM_2.5_-mediated induction of oxidative stress, subcellular organ dysfunction, and apoptosis as well as inflammation [11,12]. Among them, it was verified that oxidative stress induced by PM_2.5_ caused endoplasmic reticulum (ER) stress. Here, we further studied the contribution of PM_2.5_-induced ER stress to cell damage.

Although there is much research on the effect of atmospheric PM on normal skin, few studies have identified suitable skin protectants. Ginsenosides are the primary active ingredients of ginseng, and comprise a group containing more than 30 different triterpene saponins, the contents and relative proportions of which vary in different species of ginseng [13]. Among them, two major groups have been identified to have pharmacotherapeutic activity: panaxadiol ginsenosides such as Rb1, Rb2, Rc, and Rd, and protopanaxatriol ginsenosides such as Rg1, Re, and Rf. Studies on the effects of ginsenosides on PM_2.5_ have reported that ginsenoside Rg1 has protective effects against the toxicity of PM_2.5_ in human alveolar epithelial cells [14] and human umbilical vein endothelial cells [15]. In a previous study on skin cells, ginsenoside Rb1 reportedly inhibited UVB-induced oxidative stress [16], suppressed the expression of inflammatory factors [17], induced the expression of type I collagen [18], and played a role in wound healing [19]. Furthermore, the inhibition of apoptosis by the ginsenoside Rb1 is associated with the AhR signaling pathway [20]. Therefore, based on the above investigations, we speculated that the ginsenoside Rb1 should counteract the adverse effects of PM_2.5_ on skin cells. This study further attempted to clarify this protective mechanism. 

## 2. Materials and Methods 

### 2.1. Reagents

PM_2.5_, which is a standard diesel PM (SRM 1650b) issued by the National Institute of Standards and Technology (NIST, USA), was purchased from Sigma-Aldrich (St. Louis, MO, USA). The 1650b diesel PM, with a mean diameter of 0.18 µm, was predominantly composed of PAHs and nitro-PAHs. The certified mass fraction values of PAHs and nitro-PAHs in SRM 1650b are provided in a previous our report [12]. PM_2.5_ stock solution (25 mg/mL) was prepared in dimethyl sulfoxide (DMSO) and sonicated for 30 min to prevent the agglomeration of the suspended PM_2.5_. Experiments were performed within 1 h of stock preparation to avoid variations in the composition of PM_2.5_ in solution. Ginsenoside Rb1 was purchased from Cayman Chemical (Ann Arbor, MI, USA). 

### 2.2. Cell Culture

Two skin cell types, human HaCaT keratinocytes (CLS Cell Lines Service GmbH, Eppelheim, Germany) and normal human dermal fibroblasts (NHDF) (Lonza, Walkersville, MD, USA), were cultured in Dulbecco’s modified Eagle’s medium (DMEM) supplemented with 10% heat-inactivated fetal bovine serum (Gibco, Life Technologies, Grand Island, NY, USA) and antibiotic-antimycotic (Gibco) at 37 °C and 5% CO_2_ in a humidified atmosphere.

### 2.3. Transient Transfection with Small Interfering RNA 

HaCaT cells were seeded in 24-well plates at 0.6 × 10^5^ cells/well and transfected when they approached 50% confluence. Cells were transfected using Lipofectamine 2000 (Invitrogen, Carlsbad, CA, USA) and 10–50 nM short interfering RNA (siRNA) in accordance with the manufacturer’s instructions. The siRNA constructs targeted the following genes: mismatched siRNA control (Santa Cruz Biotechnology, Dallas, TX, USA), protein kinase R (PKR)-like endoplasmic reticulum kinase (*PERK*; Santa Cruz Biotechnology), inositol-requiring enzyme 1 (*IRE1*; Santa Cruz Biotechnology), activating transcription factor 6 (*ATF6*; Santa Cruz Biotechnology), and C/EBP homologous protein (*CHOP*; Bioneer Corporation, Daejeon, Korea). 

### 2.4. Western Blotting

HaCaT cell lysates were prepared, and total protein was extracted and separated by SDS-PAGE. The separated proteins were transferred to a nitrocellulose membrane and the membrane was incubated with the following primary antibodies: Anti-CHOP, anti-IRE1, anti-caspase-12, anti-poly (ADP-ribose) polymerase (PARP), anti-Mcl-1, anti-caspase-3, anti-caspase-9 (Cell Signaling Technology, Danvers, MA, USA), anti-phospho-IRE1 (Abcam, Cambridge, MA, USA), anti-phospho-PERK, anti-PERK, anti-phospho-eIF2, anti-glucose-regulated protein, 78 kDa (GRP78), anti-XBP-1, anti-ATF6, anti-Bcl-2, anti-Bax (Santa Cruz Biotechnology), and anti-actin (Sigma Aldrich). Subsequently, goat anti-rabbit or mouse IgG secondary antibodies conjugated with horseradish peroxidase (Invitrogen) were added and incubated. Protein expression was detected using Amersham enhanced chemiluminescence western blotting detection reagent (GE Healthcare, Buckinghamshire, UK).

### 2.5. Detection of Apoptosis

siRNA-transfected cells were stained with Hoechst 33342 cell-permeant nuclear counterstain dye (Sigma Aldrich) for 10 min, and images were acquired using a fluorescence microscope equipped with a CoolSNAP-Pro color digital camera (Media Cybernetics, Rockville, MD, USA) to identify condensed nuclei in apoptotic cells. The apoptotic index was calculated as follows: (apoptotic cell number in treated group/total cell number in treated group)/(apoptotic cell number in control group/total cell number in control group) [11].

### 2.6. Cytotoxicity Assay

siRNA (Control, PERK, IRE1, ATF6, CHOP)-transfected cells were seeded at a density of 6.0 × 10^4^ cells/well in 24-well plates and treated with PM_2.5_. Alternatively, cells were seeded in 24-well plates at a density of 6.0 × 10^4^ cells/well, treated with 10, 20, 30, 40, or 50 μM ginsenoside Rb1 for 1 h before treatment with the PM_2.5_ or ER stress-inducing agent dithiothreitol (DTT), and incubated for 24 h. The supernatant was aspirated and washed twice with PBS, followed by the addition of 3-(4,5-dimethylthiazol-2-yl)-2,5-diphenyltetrazolium bromide (MTT; Amresco Inc., Cleveland, OH, USA) solution, and was incubated for 4 h at 37 °C. The formazan crystals were dissolved in DMSO, and cell viability was evaluated by measuring the absorbance at 540 nm using a microplate reader. In addition, cells at a density of 1.0 × 10^5^ cells/well were seeded in 35-mm dishes and cultured for 16 h. Cells were pre-treated with 40 μM ginsenoside Rb1 for 1 h, and then treated with PM_2.5_ for 24 h. After adding one drop of 0.1% trypan blue solution to 0.1 mL of the cell suspension and staining at room temperature for 5 min, 10 μL of the cell suspension was used for cell enumeration using a microscope at 10× magnification to determine the number of viable and dead cells. Cell viability was calculated as follows [11]: Cell viability (%) = [unstained cells/(unstained cells + stained cells)] × 100(1)

### 2.7. Detection of ROS

Superoxide anion and hydroxyl radicals were captured with 5,5-dimethyl-1-pyrroline-N-oxide (DMPO, Cayman Chemical, Ann Arbor, MI, USA), and the obtained DMPO/•OOH [21] and DMPO/•OH adducts [22] were detected using a JES-FA200 electron spin resonance (ESR) spectrometer (JEOL, Tokyo, Japan). For the detection of cell-free ROS, superoxide anions were produced by xanthine/xanthine oxidase, and hydroxyl radicals were produced by the Fenton reaction. The set parameters were as follows: magnetic field, 336 mT; power, 5.00 mW or 1.00 mW; frequency, 9.4380 GHz; modulation width, 0.2 mT; amplitude, 500 or 100; sweep time, 0.5 min; sweep width 10 mT; time constant, 0.03 s; temperature, 25 °C. 

In addition, in order to detect intracellular ROS, cells at a density of 2.0 × 10^5^ cells/well were seeded in a 6-well plate and incubated for 16 h. After pretreatment with 40 μM ginsenoside Rb1 or positive control N-acetylcysteine (NAC) for 30 min, PM_2.5_ was added to the medium and incubated for another 30 min, then 25 μM intracellular ROS probe 2′,7′-dichlorodihydrofluorescein diacetate (H_2_DCFDA; Molecular Probes, Eugene, OR, USA) was added and reacted for 30 min. Cells were tested for ROS by BD LSRFortessa high-end performance flow cytometer (BD Biosciences, San Jose, CA, USA). For confocal microscopy, cells at a density of 6.0 × 10^4^ cells/well were seeded in 4-well chamber slides and incubated for 16 h. PM_2.5_ was added to the media after pretreatment with 40 μM ginsenoside Rb1 for 30 min. After incubation at 37 °C for 30 min, the cells were loaded with 25 μM H_2_DCFDA or 10 μM mitochondrial ROS probes dihydrorhodamine 123 (DHR 123; Molecular Probes) for 10 min; confocal microscopic images of the stained cells were assessed.

### 2.8. Single-Cell Gel Electrophoresis Assay (Comet Assay)

HaCaT and NHDF cells were coated on microscopy slides and lysed by immersing them in a buffer (2.5 M NaCl, 100 mM Na_2_EDTA, 10 mM Tris-pH 10, 1% N-lauroyl sarcosinate) in a dark room at 4 °C for 1.5 h. Electrophoresis was performed at 25 V and 300 mA for 20 min. After staining with ethidium bromide, the length of the comet tail of 50 cells on each slide was recorded using image analysis software (Kinetic Imaging, Komet 5.5, Andor, Oxford, UK) on a fluorescence microscope [11].

### 2.9. Lipid Peroxidation Assay

HaCaT and NHDF cells were stained with 5 μM fluorescent probe diphenyl-1-pyrenylphosphine (DPPP; Molecular Probes) as described previously [23] and analyzed using an Olympus FV1200 laser-scanning microscope equipped with FV10-ASW viewer 4.2 software (Tokyo, Japan).

### 2.10. Protein Carbonylation Assay

Total proteins were isolated from the cells using protein lysis buffer and quantified. The OxiSelect protein carbonyl ELISA kit (Cell Biolabs, San Diego, CA, USA) was used in accordance with the manufacturer’s instructions.

### 2.11. Quantification of Ca^2+^ Levels

HaCaT and NHDF cells were stained with 10 μM fluoro-4-acetoxymethyl ester (Fluo-4-AM) or Rhod-2 acetoxymethyl ester (Rhod-2-AM; Molecular Probes) to detect intracellular and mitochondrial Ca^2+^ levels. Fluorescence microscopic images were acquired using a confocal microscope after dye staining for 30 min and fluorescence values were measured with ImageJ software (NIH, Bethesda, MD, USA) [24].

### 2.12. Mitochondrial Membrane Potential Measurement

HaCaT and NHDF cells were stained with membrane-permeant 5,5′,6,6′-tetrachloro-1,1′,3,3′-tetraethylbenzimidazolylcarbocyanine iodide (JC-1, Invitrogen) dye for 30 min and the mitochondrial membrane potential (Δψm) was visualized via flow cytometry and confocal microscopic measurement of the ratio of red-green fluorescence [25].

### 2.13. Detection of ATP Levels

The harvested cells were washed twice with PBS and lysed using lysis buffer (100 mM Tris, 4 mM EDTA, pH 7.75) for 30 min, boiled for 2 min, and then centrifuged at 1000× *g* for 1 min. The supernatant was assayed to determine the amount of ATP using an ATP determination kit (Invitrogen) [26].

### 2.14. Statistical Analysis

All analyses were performed at least three times independently, and all data are expressed as the mean ± standard error of the mean (SEM). Statistically significant differences were determined via analysis of variance (ANOVA) and Tukey’s test for post hoc analysis using SigmaStat version 3.5 software (Systat Software Inc., San Jose, CA, USA). A value of *p* < 0.05 was considered statistically significant.

## 3. Results

### 3.1. ER Stress Mediates Apoptosis after Exposure to PM_2.5_

Because PERK, IRE1, ATF6, and CHOP are key factors associated with ER stress, their down-regulation was predicted to affect PM_2.5_-induced ER stress. To verify this, the knockdown of their genes was first confirmed by western blotting in cells transfected with siRNAs targeting PERK, IRE1, ATF6, and CHOP (Figure 1a). Using Hoechst 33342 staining, we confirmed the inhibitory effect of ER-stress-related gene down-regulation on PM_2.5_-induced apoptosis (Figure 1b). It was also confirmed by MTT assay that the down-regulation of ER stress-related genes has a protective effect on PM_2.5_-induced cells (Figure 1c).

### 3.2. Ginsenoside Rb1 Confers Protection against PM_2.5_-Induced ROS

To study the effect of ginsenoside Rb1 (Figure 2a) on PM_2.5_-induced cellular injury, we first conducted a dose-dependent toxicity test to select the optimal concentration of ginsenoside Rb1. As shown in Figure 2b, ginsenoside Rb1 showed no cytotoxicity in either HaCaT or NHDF at concentrations below 40 μM. Moreover, staining with trypan blue or MTT assay confirmed that 40 μM ginsenoside Rb1 had varying degrees of protective effects against PM_2.5_-induced cytotoxicity (Figure 2c,d). Therefore, we selected a concentration of 40 μM ginsenoside Rb1 for subsequent experiments. To determine the scavenging effect of 40 μM ginsenoside Rb1 on ROS, the scavenging effect of ginsenoside Rb1 on superoxide anion and hydroxyl radicals was first assessed by ESR spectroscopy. In the xanthine/xanthine oxidase system, the superoxide anion signal is at the 2241 signal value; however, upon ginsenoside Rb1 treatment, it was reduced to the 2024 signal value (Figure 2e). The hydroxyl radical signal generated by the Fenton reaction was also reduced by ginsenoside Rb1 from 2844 to 2065 (Figure 2f). We previously demonstrated that PM_2.5_ induces ROS [11]; therefore, the effect of ginsenoside Rb1 on PM_2.5_-induced intracellular ROS was next determined using H_2_DCFDA fluorescent dye. Flow cytometry results showed that pre-treatment with ginsenoside Rb1 or positive control NAC significantly reduced PM_2.5_-induced ROS in HaCaT and NHDF cells (Figure 2g). Confocal microscopy confirmed this result again (Figure 2h).

### 3.3. Ginsenoside Rb1 Confers Protection against PM_2.5_-Induced Oxidative Stress

Next, we evaluated the effect of ginsenoside Rb1 pre-treatment on the intracellular macromolecules damaged by PM_2.5_. Comet assays to assess DNA degradation showed that PM_2.5_ increased the concentration of DNA in the tails of cells by 30% and 22%, whereas pretreatment with ginsenoside Rb1 reduced this to 16% and 8% in HaCaT and NHDF cells, respectively (Figure 3a). In addition, determination of DPPP oxide fluorescence to identify lipid peroxidation was performed in PM_2.5_-treated cells, and fluorescence was significantly reduced after pre-treatment with ginsenoside Rb1 (Figure 3b). Similarly, the level of carbonylation in biomarkers of oxidative damage proteins in PM_2.5_-treated cells was relatively high, and ginsenoside Rb1 prevented this PM_2.5_-induced carbonyl formation (Figure 3c). Taken together, these results indicate that ginsenoside Rb1 protects intracellular components from damage by inhibiting oxidative stress caused by PM_2.5_-induced ROS.

### 3.4. Ginsenoside Rb1 Confers Protection against PM_2.5_-Induced ER Stress

It was previously demonstrated that PM_2.5_-induced oxidative damage leads to ER stress [11]. Therefore, we next investigated whether the cytoprotective effect of ginsenoside Rb1 on PM_2.5_ is related to blocking ER stress-related pathways. ER is the predominant intracellular Ca^2+^ reservoir, and the disruption of Ca^2+^ homeostasis activates ER stress [27,28]. Confocal microscopy analysis showed that the Ca^2+^ fluorescence of cells treated with PM_2.5_ was stronger than that of the control group; this fluorescence was significantly attenuated upon pre-treatment with ginsenoside Rb1 (Figure 4a). ER stress is mediated by three ER sensors: PERK, IRE1, and ATF6 [29]. GRP78, as a key element, maintains normal ER function in cells and protects the ER from dangerous stimuli. CHOP is a transcription factor that plays an important role in promoting ER stress-induced apoptosis [30]. To further investigate the effect of ginsenoside Rb1 on PM_2.5_-induced ER stress, we measured the protein expression of GRP78 and CHOP. Western blot analysis revealed that GRP78 and CHOP expression was increased after PM_2.5_ treatment, and was significantly inhibited after pre-treatment with ginsenoside Rb1 (Figure 4b). Similarly, ginsenoside Rb1 pre-treatment inhibited the PM_2.5_-induced expression of ER stress-related proteins such as phospho-PERK, phospho-eIF2, phospho-IRE1, XBP-1, active ATF6 and active caspase-12 (Figure 4c).

### 3.5. Ginsenoside Rb1 Protects against PM_2.5_-Induced Mitochondrial Damage

Confocal microscopy analysis showed that PM_2.5_ treatment increased ROS production and Ca^2+^ overload in mitochondria, both of which were inhibited by pre-treatment with ginsenoside Rb1 (Figure 5a,b). The permeability of mitochondrial membranes associated with apoptosis is mediated by the release of cytochrome c and caspase activation [31]. To determine the change in Δψm, which indicates mitochondria-dependent apoptosis, we used the cationic lipophilic fluorescent dye JC-1. Flow cytometry analysis showed that the JC-1 red/green fluorescence ratio decreased in PM_2.5_-treated cells, indicating the depolarization of Δψm, whereas pre-treatment with ginsenoside Rb1 reversed this result (Figure 5c). Confocal microscopy images also showed consistent results (Figure 5d). Furthermore, the amount of ATP in the cells was measured by the ATP determination kit. Treatment with PM_2.5_ was found to reduce index of ATP level to 0.795 and 0.233 in HaCaT and NHDF cells, respectively; however, pretreatment with ginsenoside Rb1 restored index of ATP level to 0.94 and 0.283 in both cells, respectively (Figure 5e).

### 3.6. Ginsenoside Rb1 Protects against PM_2.5_-Induced Apoptotic Cell Death

We have previously demonstrated that PM_2.5_ can induce apoptosis, which was confirmed by Hoechst 33342 staining [11]; pre-treatment with ginsenoside Rb1 was found to significantly inhibit the PM_2.5_-induced formation of apoptotic bodies (Figure 6a). As shown in Figure 6b, pre-treatment with ginsenoside Rb1 inhibited the PM_2.5_-induced expression of the pro-apoptotic protein Bax, and partially restored the expression of the anti-apoptotic proteins Bcl-2 and Mcl-1. We also found that ginsenoside Rb1 inhibited PM_2.5_-induced expression of active caspase-9 and active caspase-3. These observations were confirmed based on the active form of PARP by the active caspases (Figure 6c).

### 3.7. Ginsenoside Rb1 Protects against Mitochondria-Associated ER Stress-Mediated Apoptosis

To confirm these findings, we used ER stress inducers to examine the effects of ginsenoside Rb1 on ER stress in HaCaT and NHDF cells. Fluorescence microscopic cell images of Hoechst 33342-stained nuclei showed that ginsenoside Rb1 significantly reduced the proportion of apoptotic bodies induced by the ER stress inducer DTT (Figure 7a). The MTT assay also confirmed the significant protective effect of ginsenoside Rb1 against DTT-induced cell death (Figure 7b). To determine the correlation between mitochondria and ER during the process of cell damage induced by PM_2.5_, ATP levels, apoptosis, and cell viability were measured after treatment with the ER stress inhibitor tauroursodeoxycholic acid (TUDCA). The decrease in ATP levels induced by PM_2.5_ was partially recovered by treatment with TUDCA or ginsenoside Rb1, and the ATP recovery rate in the PM_2.5_ group co-pre-treated with TUDCA and ginsenoside Rb1 was significantly higher than that in the PM_2.5_ group pre-treated with ginsenoside Rb1. However, there was no significant difference between the ATP recovery rate of the PM_2.5_ group co-pre-treated with TUDCA and ginsenoside Rb1 and the PM_2.5_ group pre-treated with only TUDCA (Figure 7c). After treatment with TUDCA, the results of apoptosis detected using Hoechst 33342 staining (Figure 7d) and the results of cell viability detected with trypan blue reagent (Figure 7e) and MTT assay (Figure 7f) also showed similar protection patterns. These results indicate that cell damage caused by PM_2.5_ is not only directly caused by induced ER stress, but also by ER stress-associated mitochondrial dysfunction, which is inhibited by ginsenoside Rb1.

## 4. Discussion

In recent years, many papers have described the association between ER stress and apoptosis [32,33]. We have also emphasized the contribution of ER stress to PM_2.5_-induced apoptosis [11]. Here, by down-regulating several genes related to ER stress, such as *PERK*, *IRE1*, *ATF6,* and *CHOP*, we further demonstrated that ER stress played an important role in the process of apoptosis induced by PM_2.5_. 

As the concentration of PM_2.5_ in the air is increasing daily, there is an urgent need to develop appropriate countermeasures to PM_2.5_-induced damage. Rather than focusing on ways to remove the causes of PM_2.5_ exposure, our first priority should be to prevent the detrimental effects of PM_2.5_ on health. Therefore, we are committed to exploring bioactive substances that are resistant to PM_2.5_ and to studying their mechanisms. Ginsenosides have long been known to increase immunity, relieve fatigue, and eliminate toxins. However, among the more than 30 types of ginsenosides, only ginsenoside Rg1 has been tested for its effectiveness against PM_2.5_ [14,15], and whether ginsenoside Rb1 has a protective effect on skin cells exposed to PM_2.5_ has not been studied until now.

In the present study, the cytotoxicity of ginsenoside Rb1 was first determined by an MTT assay, and 40 μM ginsenoside Rb1, which was determined as optimal (Figure 2c,d) and resulted in a survival rate higher than 95%, was selected to verify its protective effect against PM_2.5_-induced cytotoxicity using the trypan blue assay and MTT assay. We then examined the ROS scavenging rate of ginsenoside Rb1 using the ESR method and demonstrated that ginsenoside Rb1 itself has superoxide anion and hydroxyl radical scavenging ability. It also showed an ability to scavenge PM_2.5_-induced ROS within cells (Figure 2g,h). Oxidative stress is caused by the overproduction and accumulation of ROS, which overwhelms the already compromised antioxidant defense mechanism, which is a form of cellular injury. Next, we determined whether ginsenoside Rb1 is resistant to PM_2.5_-induced oxidative stress and found that it inhibits the oxidation of macromolecular substances and prevents DNA damage, lipid peroxidation, and protein carbonylation (Figure 3).

In the pathophysiology of many diseases, ER stress is known to be correlated with oxidative stress. Studies show that oxidative stress affects the protein folding pathway, leading to reduced efficiency and increased misfolded protein production, resulting in a redox imbalance that can exacerbate ER stress [34]. The disruption of intracellular Ca^2+^ homeostasis is known to be closely related to ER stress. Our results showed that ginsenoside Rb1 inhibits PM_2.5_-induced ER stress and inhibits the increase in intracellular Ca^2+^ levels associated with this stress. ER stress induces a high expression of CHOP, a member of the bZIP transcription factor C/EBP family. Furthermore, GRP78 is a major ER chaperone protein that plays an important role in protein quality control in the ER, as well as in the regulation of three ER transmembrane proteins: PERK, IRE1α, and ATF6 [35,36]. Here, we demonstrated that ginsenoside Rb1 significantly inhibits the expression of PM_2.5_-induced ER stress-related proteins. 

ROS overproduction causes oxidative stress, which leads to the rapid depolarization of mitochondrial membrane potential, subsequently impairing oxidative phosphorylation. By activating the inter-mitochondrial signaling network, ROS levels, particularly superoxide anions and hydrogen peroxide, are increased by the damaged mitochondria, which enhances the propagation of mitochondrial-driven ROS [37]. Calcium signaling in mitochondria is an important feature of cell survival, as it affects cellular energy production and determines the cell fate in terms of inducing and preventing apoptosis [38]. Ginsenoside Rb1 effectively inhibited the PM_2.5_-induced production of ROS and Ca^2+^ in mitochondria (Figure 5a,b). ER stress-induced apoptosis mainly involves multiple pathways, such as the caspase family, CHOP, the Bcl-2 family, and Ca^2+^ toxicity. As in mitochondria, the Bcl-2/Bax protein family is also present in the ER and regulates the Ca^2+^ balance in the ER, controls ER stress inducers, and peroxide-induced cell death [39]. The anti-apoptotic protein Mcl-1 and the pro-apoptotic protein Bik are mainly located in the ER [40]. Bcl-2 was found to be the target gene of CHOP and to be down-regulated by its activity [41]. The regulation of apoptosis by Bcl-2 family proteins is controlled by mitochondrial permeability. Apoptosis induced by PM_2.5_ in normal human keratinocytes is, to a large extent, a result of the inhibitory activity of Bcl-2/Mcl-1. Pre-treatment with ginsenoside Rb1 in this study largely prevented this decrease and partially restored the expression of the pro-apoptotic protein Bax, which was inhibited by PM_2.5_ (Figure 6b). Caspase-9 is reportedly activated after the disruption of the mitochondrial membrane. The caspase-9 signaling cascade induces mitochondrial feedback disruption by cleaving the anti-apoptotic proteins Bcl-2, Bcl-xL, and Mcl-1 [42]. Therefore, the active form of caspase-9 and its target, caspase-3, along with their substrate PARP, were evaluated, as shown by western blot analysis. We have demonstrated that ginsenoside Rb1 can alter the expression or activation of these key regulatory proteins in the caspase pathway (Figure 6c). 

Although these results indicate that ginsenoside Rb1 inhibits mitochondria-mediated caspase-dependent apoptosis, ginsenoside Rb1 was also found to inhibit ER stress-induced apoptosis and cell death. Increasing evidence clearly shows that ER function is closely related to mitochondrial function, that Ca^2+^ signaling is a major hub of this intracellular interaction, and that the ER and mitochondria cooperate to stimulate cell death [43]. We confirmed that PM_2.5_ disrupted the balance of Ca^2+^ in the intracellular space and in the mitochondria, and that this was reversed by ginsenoside Rb1 (Figure 4a and Figure 5b). We have also demonstrated that cell damage caused by PM_2.5_ is not only directly induced by ER stress, but also by ER stress associated with the mitochondrial pathway (Figure 7c). 

## 5. Conclusions

In conclusion, our study showed that ER stress plays an important role in PM_2.5_-induced cell damage, and that ginsenoside Rb1 has a significant protective effect against this PM_2.5_-induced cell damage (Figure 8). Ginsenoside Rb1 not only inhibits mitochondria-dependent apoptosis caused by PM_2.5_-induced oxidative stress, but also plays a cytoprotective role by inhibiting ER stress. The cell damage process of PM_2.5_ appears to follow the ER stress or mitochondrial dysfunction pathway, but is actually a complex process involving both ER stress and the mitochondrial pathway. Finally, ginsenoside Rb1 showed excellent inhibitory activity throughout the entire process of cell damage by PM_2.5_. The 50 μg/mL of PM_2.5_ used in our system is the dose that ultimately led to cell death after a series of changes in the cells. We believe that it is unlikely that one-time exposure of such a dose will occur in people’s daily lives, but long-term exposure causes PM_2.5_ to accumulate in the body, causing similar adverse effects. We can thus conclude that ginsenoside Rb1 can be used as a suitable raw material to develop efficient skin care products to prevent harm from PM_2.5_, and thus, this study provides valuable information that could help to overcome the threat posed by PM_2.5_ to human health.

## Figures and Tables

**Figure 1 antioxidants-08-00383-f001:**
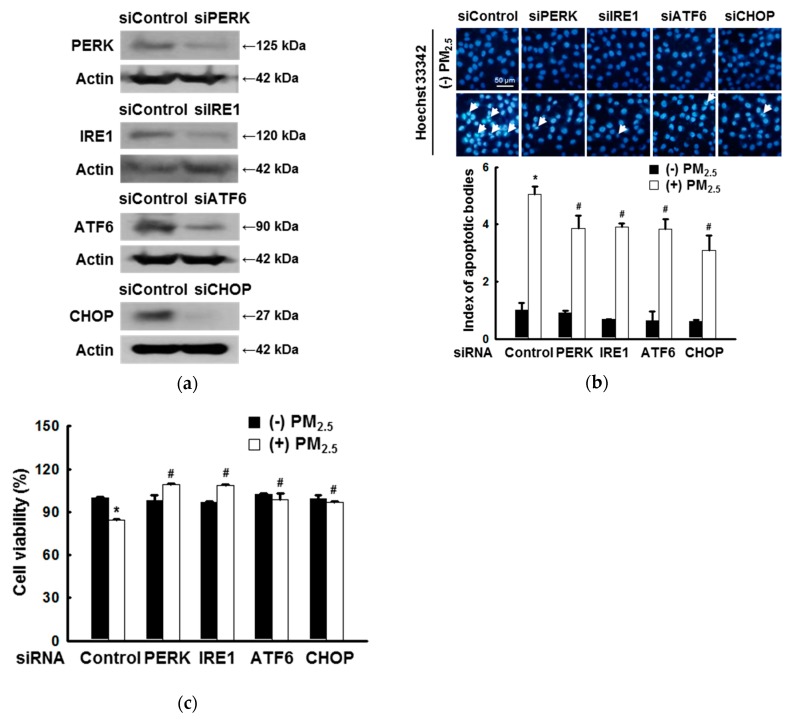
Endoplasmic reticulum (ER) stress mediates apoptosis after exposure to particulate matter 2.5 (PM_2.5_). (**a**) HaCaT cells were transfected with control siRNA and siRNAs against *PERK*, *IRE1*, *ATF6*, and *CHOP* as verified by western blotting. The control, PERK, IRE1, ATF6, and CHOP siRNA-transfected cells were treated with PM_2.5_. (**b**) Nuclei were stained with Hoechst 33342 and images were acquired using a fluorescence microscope. Arrows indicate apoptotic cells. (**c**) Cell viability was assessed using MTT assay. (**b**,**c**) * *p* < 0.05 compared to siControl; ^#^
*p* < 0.05 compared to siControl + PM_2.5_.

**Figure 2 antioxidants-08-00383-f002:**
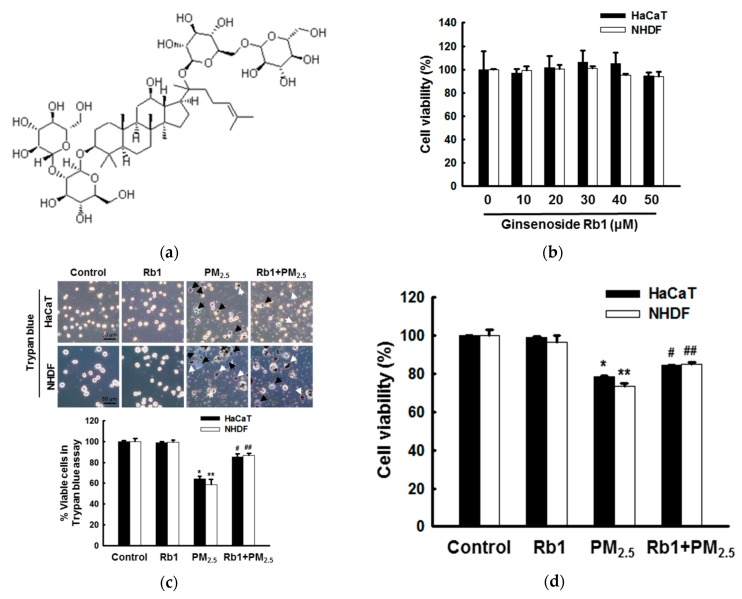
Ginsenoside Rb1 confers protection from PM_2.5_-induced intracellular reactive oxygen species (ROS). (**a**) Chemical structure of ginsenoside Rb1. (**b**) Cells were seeded, and ginsenoside Rb1 was added at final concentrations of 10, 20, 30, 40, and 50 μM. After 24 h, cell viability was determined using the MTT assay. (**c**,**d**) Cells were pre-treated with ginsenoside Rb1 (40 μM) for 1 h, treated with PM_2.5_ (50 μg/mL), incubated for 24 h, (**c**) stained with trypan blue reagent, and visualized using a phase contrast microscope to evaluate cell viability. Black arrows indicate dead cells and white arrows indicate PM_2.5_. (**d**) Cell viability was assessed using MTT assay. (**e**) Superoxide anions generated by the xanthine/xanthine oxidase system were reacted with DMPO and the resultant DMPO/·OOH adducts were detected using ESR spectrometry. * *p* < 0.05 compared to the control; ^#^
*p* < 0.05 compared to superoxide anions. (**f**) Hydroxyl radicals generated by the Fenton reaction (H_2_O_2_+FeSO_4_) were reacted with DMPO and the resultant DMPO/·OH adducts were detected by ESR spectrometry. * *p* < 0.05 compared to the control; ^#^
*p* < 0.05 compared to hydroxyl radicals. (**g**,**h**) Cells were pre-treated with ginsenoside Rb1 or NAC for 1 h and then treated with PM_2.5_ for the indicated time periods. ROS levels were assessed by (**g**) flow cytometry and (**h**) confocal microscopy after H_2_DCFDA staining (FI: Fluorescence intensity). (**c**,**d**,**h**) *,** *p* < 0.05 compared to the control groups of both HaCaT and NHDF cells, respectively; ^#^,^##^
*p* < 0.05 compared to PM_2.5_-treated groups of both HaCaT and NHDF cells, respectively.

**Figure 3 antioxidants-08-00383-f003:**
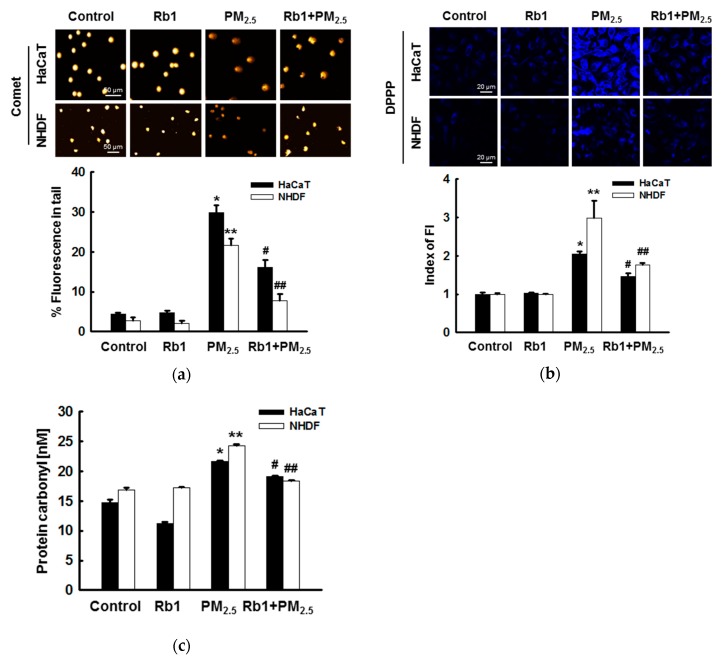
Ginsenoside Rb1 confers protection from PM_2.5_-induced oxidative stress. (**a**) DNA damage was evaluated by the comet assay; representative images show comet tails, and the graph shows the quantification of cellular DNA damage. (**b**) Lipid peroxidation was analyzed by confocal microscopy after DPPP staining. (**c**) Protein oxidation was assayed by measuring carbonyl formation. (**a**–**c**) *,** *p* < 0.05 compared to the control groups of both HaCaT and NHDF cells, respectively; ^#^,^##^
*p* < 0.05 compared to PM_2.5_-treated groups of both HaCaT and NHDF cells, respectively.

**Figure 4 antioxidants-08-00383-f004:**
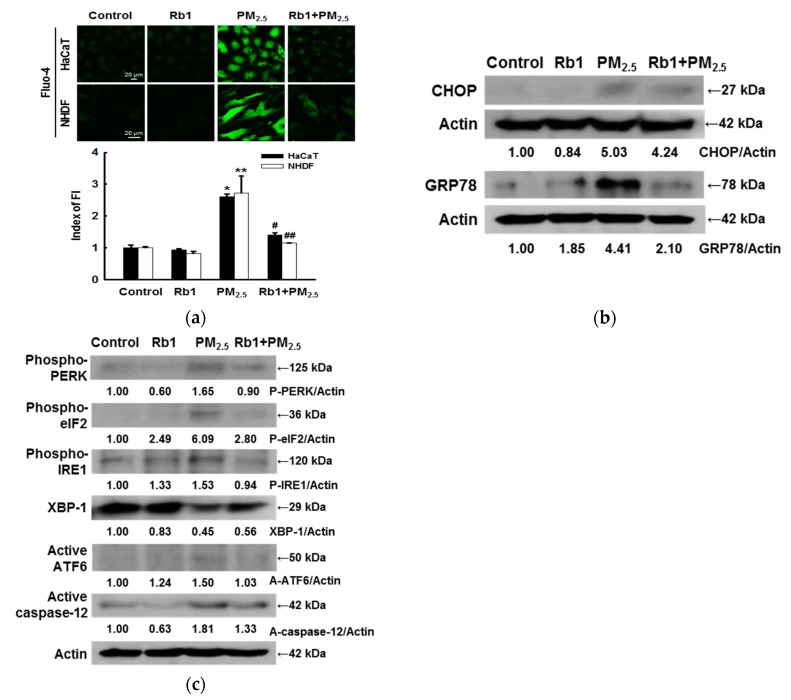
Ginsenoside Rb1 confers protection from PM_2.5_-induced ER stress. Cells were pre-treated with ginsenoside Rb1 for 1 h, followed by treatment with PM_2.5_ (50 μg/mL) for 24 h. (**a**) Intracellular Ca^2+^ levels were examined by confocal microscopy using Fluo-4-AM (FI: Fluorescence intensity). *,** *p* < 0.05 compared to the control groups of both HaCaT and NHDF cells, respectively; ^#^,^##^
*p* < 0.05 compared to PM_2.5_-treated groups of both HaCaT and NHDF cells, respectively. In HaCaT cells, the expression of the proteins (**b**) CHOP, GRP78, (**c**) phospho-PERK, phospho-eIF2, phospho-IRE1, XBP-1, active ATF6 and active caspase-12 were analyzed by western blotting.

**Figure 5 antioxidants-08-00383-f005:**
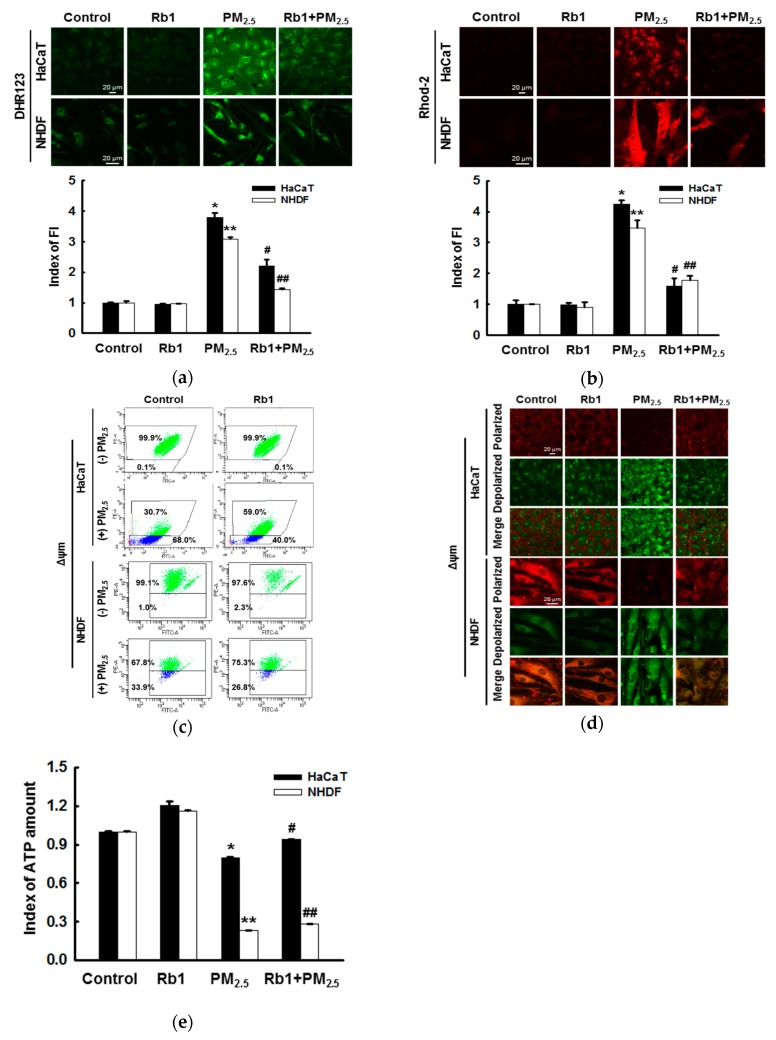
Ginsenoside Rb1 confers protection against PM_2.5_-induced mitochondrial damage. Cells were pre-treated with ginsenoside Rb1 for 1 h, followed by treatment with PM_2.5_ (50 μg/mL) for 24 h; they were then analyzed by confocal microscopy to assess (**a**) mitochondrial ROS (DHR 123 staining) and (**b**) mitochondrial Ca^2+^ levels (Rhod-2 AM staining). FI: Fluorescence intensity. The Δψm was determined using JC-1 dye by (**c**) flow cytometry and (**d**) confocal microscopy. (**e**) The amount of ATP in the cells was measured using an ATP determination kit. (**a**,**b**,**e**). *,** *p* < 0.05 compared to the control groups of both HaCaT and NHDF cells, respectively; ^#^,^##^
*p* < 0.05 compared to PM_2.5_-treated groups of both HaCaT and NHDF cells, respectively.

**Figure 6 antioxidants-08-00383-f006:**
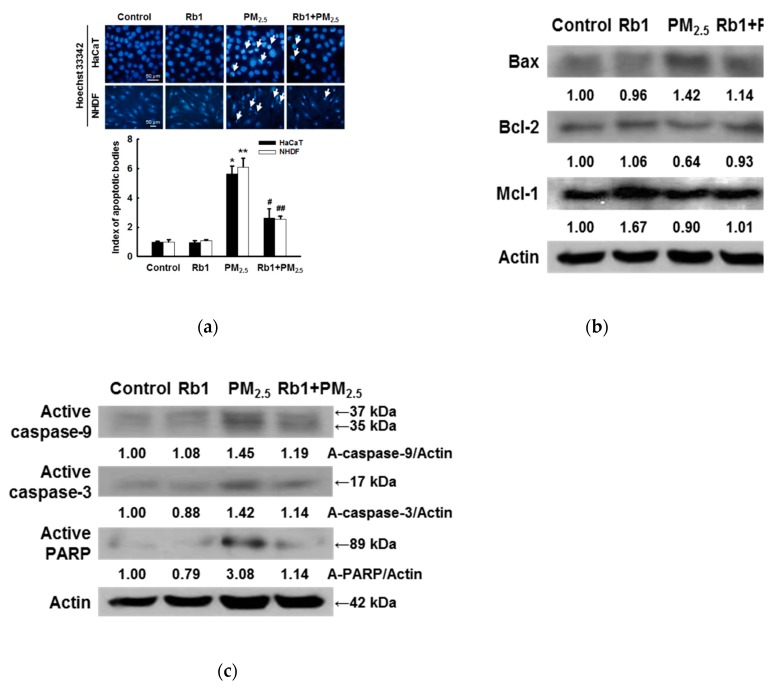
Ginsenoside Rb1 confers protection against PM_2.5_-induced apoptosis. (**a**) Cells were pre-treated with ginsenoside Rb1 for 1 h, followed by treatment with PM_2.5_ (50 μg/mL) for 24 h, and then analyzed for apoptotic body formation by Hoechst 33342 staining. Apoptotic bodies are indicated by arrows. *,** *p* < 0.05 compared to the control groups of both HaCaT and NHDF cells, respectively; ^#^,^##^
*p* < 0.05 compared to PM_2.5_-treated groups of both HaCaT and NHDF cells, respectively. (**b**) HaCaT cell lysates were analyzed for the expression of Bax, Bcl-2, and Mcl-1 by western blotting; actin was used as a loading control. (**c**) The expression of caspase-3, caspase-9, and PARP was also analyzed; actin was used as a loading control.

**Figure 7 antioxidants-08-00383-f007:**
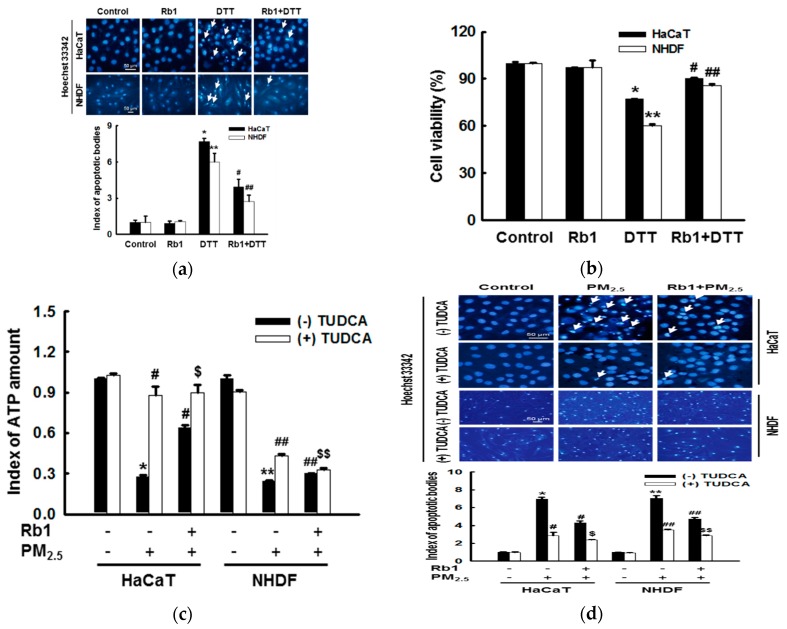
Ginsenoside Rb1 confers protection against ER stress-induced apoptosis. (**a,b**) Cells were pretreated with ginsenoside Rb1 for 1 h, and then treated with the ER inducer DTT for 24 h. (**a**) Apoptosis was detected with Hoechst 33342 dye, and (**c**) the cell viability was measured using the MTT assay. (**a,b**). *,** *p* < 0.05 compared to the control groups of both HaCaT and NHDF cells, respectively; ^#^,^##^
*p* < 0.05 compared to DTT-treated groups of both HaCaT and NHDF cells, respectively. (**c**–**e**) Cells were pre-treated with ginsenoside Rb1 (40 μM) or TUDCA (1 μM) for 1 h, then treated with PM_2.5_ (50 μg/mL) and incubated for 24 h. (**c**) The amount of ATP in the cells was measured using an ATP determination kit. (**d**) The cells were stained with Hoechst 33342 and images were acquired using a fluorescence microscope to detect apoptosis. Arrows indicate apoptotic cells. (**e**) Cells were stained with trypan blue reagent and images were taken using a phase contrast microscope to detect cell viability. Black arrows indicate dead cells and white arrows indicate PM_2.5_. (**f**) Cell viability was assessed using MTT assay. (**c**–**f**) *,** *p* < 0.05 compared to the control groups of both HaCaT and NHDF cells, respectively; ^#^,^##^
*p* < 0.05 compared to PM_2.5_-treated groups of both HaCaT and NHDF cells, respectively. ^$^,^$$^
*p* < 0.05 compared to ginsenoside Rb1 + PM_2.5_ groups of both HaCaT and NHDF cells, respectively.

**Figure 8 antioxidants-08-00383-f008:**
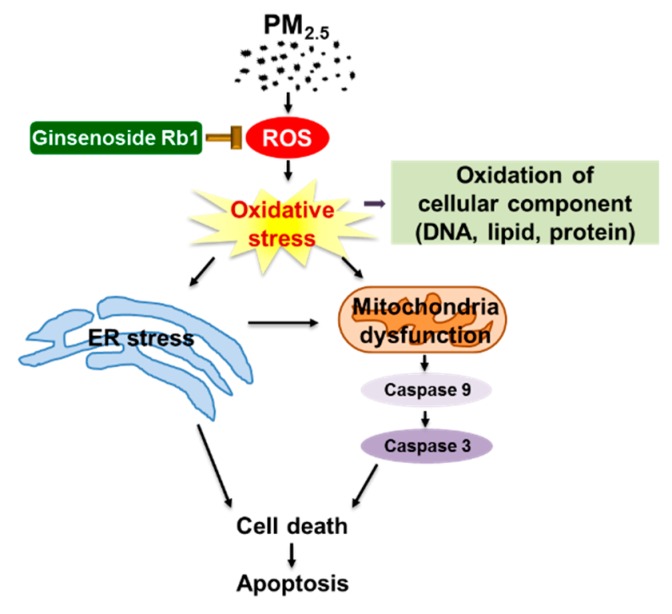
A schematic diagram of the protective effect of ginsenoside Rb1 against PM_2.5_. Ginsenoside Rb1 inhibits PM_2.5_-induced oxidative stress and prevents ER stress generation and mitochondrial dysfunction, thereby protecting cells from PM_2.5_ injury.

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
