# Peer review of "Particulate Matter 2.5 Mediates Cutaneous Cellular Injury by Inducing Mitochondria-Associated Endoplasmic Reticulum Stress: Protective Effects of Ginsenoside Rb1"

_antioxidants, 2019, doi:10.3390/antiox8090383_

Round 1
Reviewer 1 Report
In this paper Authors report the results of the studies undertaken to elucidate the mechanism by which particulate matter (PM2.5) induces damage in human HaCaT keratinocytes and normal human dermal fibroblasts, and to evaluate the preventive capacity of the ginsenoside Rb1.
In this paper Authors report the results of the studies undertaken to elucidate the mechanism by which particulate matter (PM2.5) induces damage in human HaCaT keratinocytes and normal human dermal fibroblasts, and to evaluate the preventive capacity of the ginsenoside Rb1.
Actually they show by means of appropriate methods and convincing results that PM2.5 induce oxidative stress by increasing the production of reactive oxygen species, leading to DNA damage, lipid peroxidation, and protein carbonylation via endoplasmic reticulum (ER) stress, apoptosis and cell death.
It was also clear that this effect was inhibited by ginsenoside Rb1.
Therefore, the results seem to confirm the rationale and on this base, authors suggest that ginsenoside Rb1 could be used in skin care products to protect the skin against damage by fine particles.
Despite the good work, both induction of apoptosis by PM2.5 and the protective effect of ginsenoside Rb1 have been already reported in other cell models .
In addition, some points need to be more explained and/or clarified.
About the results depicted in Figure 4e, they state that the reduction of PM2.5-induced ATP levels was largely restored by treatment with ginsenoside Rb1. However, it appears not so large, but this effect of Rb1 occurred to a lesser extent than the others reported in the other figures.
In addition, in Figure 6b they have measured cell viability by MTT after treatment with DTT.
I wonder whether MTT is the proper method to determine cytotoxicity by DTT since this is a reducing agent and MTT method is based on the reduction of tetrazolium salt by cellular dehydrogenases. Thus, did the authors verify any possible interference between the redox reactions DTT-MTT and MTT-dehydrogenases??
Besides, why the treatment with DTT is the only reported in the section “2.6. Cytotoxicity Assay” ??
Author Response
Please see attachfile

Reviewer 2 Report
The manuscript by Piao et al. describes the potential use of Ginsenoside Rb1 to counteract the oxidative effects of particulate matter 2.5 (PM2.5) within the framework of cutaneous injury. PM2.5 was demonstrated to induce a variety of oxidative stress-related outcomes (e.g. overproduction of ROS, ER stress, lipid peroxidation, DNA damage, protein carbonylation, apoptosis, etc) to human HaCaT keratinocytes and normal human dermal fibroblast (NHDF). However, Ginsenoside Rb1 was shown to partially attenuate the aforementioned PM2.5 induced effects trough its antioxidant activity. Overall, the manuscript is well-written and organized. The biochemical analysis conducted in this work is comprehensive and insightful. I would therefore like to recommend its publication in Antioxidants, pending the address of the following comments:
Can the authors comment on the dose of PM 2.5 as opposed to the realistic exposure level of PM2.5 an individual would be subjected to? The characterization data (e.g. hydrodynamic size measurement, zeta potential, particle morphology, etc) of the PM2.5 used in this study is missing. This is a major flaw of this study. Figure 1, 5 and 6: The use of Hoechst 33342 staining as a measurement of cell apoptosis is an indirect approach. Authors should supplement it with MTT cytotoxic assays to support their data. Figure 2: Ginsenoside Rb1 is proposed to function as a ROS scavenger. Authors should benchmark the antioxidant activity of ginsenoside Rb1 to that of the classical antioxidants such as N-Acetylcysteine, ascorbic acid, etc. Minor comment: scale bars are missing for figures 1b, 2c, 2g, 5a, 6a, 6d, 6eAuthor Response
Please see attachfile.

Round 2
Reviewer 1 Report
Authors have satisfactorily replied to the questions raised by this reviewer.
Reviewer 2 Report
The authors have responded satisfactorily to the comments that were raised by the reviewer.